# Construction of Local Drug Delivery System on Titanium-Based Implants to Improve Osseointegration

**DOI:** 10.3390/pharmaceutics14051069

**Published:** 2022-05-17

**Authors:** Fanying Meng, Zhifeng Yin, Xiaoxiang Ren, Zhen Geng, Jiacan Su

**Affiliations:** 1Institute of Translational Medicine, Shanghai University, Shanghai 200444, China; mfy@shu.edu.cn; 2School of Medicine, Shanghai University, Shanghai 200444, China; 3School of Life Sciences, Shanghai University, Shanghai 200444, China; 4Department of Orthopedics, Shanghai Zhongye Hospital, Shanghai 200941, China; dryzf2017@163.com

**Keywords:** local drug delivery system, titanium, bone healing, osseointegration, surface modification

## Abstract

Titanium and its alloys are the most widely applied orthopedic and dental implant materials due to their high biocompatibility, superior corrosion resistance, and outstanding mechanical properties. However, the lack of superior osseointegration remains the main obstacle to successful implantation. Previous traditional surface modification methods of titanium-based implants cannot fully meet the clinical needs of osseointegration. The construction of local drug delivery systems (e.g., antimicrobial drug delivery systems, anti-bone resorption drug delivery systems, etc.) on titanium-based implants has been proved to be an effective strategy to improve osseointegration. Meanwhile, these drug delivery systems can also be combined with traditional surface modification methods, such as anodic oxidation, acid etching, surface coating technology, etc., to achieve desirable and enhanced osseointegration. In this paper, we review the research progress of different local drug delivery systems using titanium-based implants and provide a theoretical basis for further research on drug delivery systems to promote bone–implant integration in the future.

## 1. Introduction

Titanium (Ti) and its alloys are the primary materials for orthopedic and dental implants because of their high corrosion resistance, good biocompatibility, and excellent mechanical properties [1,2,3]. Clinically, osseointegration is a vital prerequisite for the successful fixation of implants in patients [4,5,6]. The concept of osseointegration was first put forward by Branemark et al. in the late 1960s. It is defined as the direct and orderly structural and functional connection between living bone and loaded implant [7]. Although Ti-based implants are the gold standard of clinical implants, the lack of bone–implant integration is still the leading reason that hinders the success of operations [8,9]. In severe cases, poor osseointegration may give rise to a second surgery or even death, causing great physical and psychological harm to patients [10,11]. Therefore, there is an urgent need to improve the osseointegration of Ti-based implants to meet clinical needs.

Previous studies have pointed out that the biological and physicochemical properties of the implant surface have significant influences on the speed, quality, and quantity of osseointegration [12,13]. Therefore, surface modification of implants is an effective method to improve bone–implant integration [14,15]. The commonly used surface modification methods can be divided into the additive modification and subtractive modification [16]. Additive modification refers to the addition of extra materials to implants, including inorganic/organic coating, growth factor, active ion, etc., in order to enhance the bioactivity of implants [17,18,19]. Subtractive modification methods (including anodic oxidation, sandblasting, laser treatment, acid–alkali treatment, etc.) can be interpreted as the formation of rough micro/nanostructure on implants to induce the adhesion, proliferation, and differentiation of osteoblasts [20,21]. Previous studies have shown that the main causes of poor osseointegration include biologically inert, inferior antibacterial ability, and easily induced inflammatory reaction [22,23,24,25,26]. However, these traditional methods cannot completely overcome the above factors and exhibit satisfactory osseointegration.

Many studies have suggested that drug-assisted therapy can improve the osseointegration of implants [27]. Many drugs, including synthetic metabolic drugs, anticatabolic drugs, antimicrobials, and anti-inflammatory drugs are proven to remarkably optimize osseointegration [28,29]. For example, synthetic metabolic drugs could enhance osseointegration by accelerating bone deposition around the implant [30]. Antimicrobials could improve osseointegration by inhibiting infection [31]. Generally, the research and application of these drugs were based on conventional systemic therapy. However, this therapy still had several side effects, such as high biological toxicity, short duration, low targeting, etc. [32,33,34]. Recently, with the gradual development of the biomedical field, drugs can be loaded on the implant surface to build a local drug delivery system [35,36,37]. This system avoids the deficiency of systemic drug delivery, but also promotes bone formation, as well as antibacterial and anti-inflammatory effects [38]. Therefore, constructing a local drug delivery system that uses Ti-based implants is a promising method to achieve ideal osseointegration.

Herein, we review the research progress of the local drug delivery system using Ti-based implants (Figure 1). The construction methods and the possible effects of local delivery of different drugs on osseointegration are discussed. We hope this review can provide a theoretical basis for the clinical optimization of bone–implant integration.

## 2. Local Drug Delivery Systems with Ti-Based Implants

### 2.1. Construction Approaches of Local Drug Delivery Systems with Ti-Based Implants

In order to realize controlled drug delivery, it is necessary to fabricate appropriate drug delivery systems with Ti-based implants. At present, the main approaches to constructing drug delivery systems with Ti-based implants include electrochemical anodization, sandblasting and acid etching (SLA), dopamine (DA) immobilization, and layer-by-layer (LBL) self-assembly. The advantages and limitations of the above methods are presented in Table 1.

*Electrochemical anodization*: Electrochemical anodization is a strategy of forming an oxide film on the surface of metals and their alloys [39,40]. This method is usually used to fabricate TiO_2_ nanotubes (TNTs) when constructing a drug delivery system [41]. TNTs are arranged vertically on Ti substrates to simulate the nanostructures in natural tissues [42]. On the one hand, the tubular diameters of TNT can be adjusted by changing the voltage and pH in the anodizing process to obtain a suitable tubular structure to load and deliver drugs [43]. On the other hand, TNTs prepared by anodization have been proved to regulate the behavior of osteoblasts and stem cells and effectively improve osseointegration [44].

*SLA*: Currently, SLA is the most commonly used strategy for surface modification of implants [45]. This strategy means that the abrasive medium material is sprayed on the surface of the implant by high-speed air flow to form a depression [46,47]. After that, acid etching is used to form smaller secondary structures and to clean impurities on the implant surface [48]. SLA can increase the roughness of implants, facilitate drug loading, and accelerate new bone formation around the implant [49].

*DA immobilization*: DA immobilization refers to the loading of drugs or factors on the Ti-based implants with the assistance of DA [50]. On the one hand, the chemical composition of DA is similar to that of mussel adhesion proteins, which have strong adhesion and can stabilize drugs or other bioactive molecules [51]. On the other hand, DA has excellent biocompatibility and biodegradability in vivo [52].

*LBL self-assembly*: LBL self-assembly is a surface modification method based on the alternating assembly of oppositely charged polyelectrolytes to fabricate multilayer coatings [53,54]. This method is easy to control the thickness of coatings but also can release drugs layer by layer to promote osseointegration [55].

### 2.2. Antimicrobial Drug Delivery System

According to the survey, implant-related infections occur in approximately 5–10% of orthopedic patients [56]. The infection is mainly due to the adsorption of bacteria on implants and the formation of bacterial biofilms [57]. The bacterial biofilm enhances the resistance of bacteria to the immune system and antibiotics [58,59,60]. At present, the most commonly applied antibacterial treatment is systemic injection or oral antibiotics. Nevertheless, their limitations (including low local concentration, low targeting, and drug resistance to traditional systemic therapy) pose great challenges to clinical treatment [61,62,63]. The researchers found that the local drug delivery system constructed on the implant surface has a high drug loading surface area and low drug delivery kinetics, which is expected to overcome the limitations of traditional systemic therapy [64]. The following is an overview of recent studies on the construction of Ti-based implants in a variety of ways for local delivery of different antimicrobials.

#### 2.2.1. Vancomycin

Vancomycin (Van) is a glycopeptide antibiotic. It has a good antibacterial activity for most Gram-positive bacteria due to the inhibition of the growth and reproduction of bacteria [65,66]. Therefore, Van is widely used to promote implant antibacterial activity and osseointegration capability [67]. For example, Zhang et al. demonstrated that Van-loaded TNTs had increased antibacterial activity both in vitro and in vivo and did not weaken the function of osteoblasts [68]. Moreover, several experiments showed that the construction of different coatings on Ti-based implants was more beneficial to the sustained release, antibacterial, and osteogenesis of the drug in vivo. For instance, Yuan et al. fabricated Van-loaded Ti-based implants with a multilayer, functional polymer coating. The implant could not only slowly release Van through the hyaluronidase degradation of the coating but also improve osseointegration via inhibiting the attachment of bacteria and promoting the attachment of osteoblasts. (Figure 2A) [69]. In addition, researchers constructed a drug delivery system (Van-loaded TNTs) with silk fibroin coating. Silk fibroin is a commonly used biological coating because of its slow degradation rate and excellent biological properties [70,71]. Fathi et al. confirmed that the system enables the continuous release of Van and the formation of bacterial biofilm and also promotes Ti implant osseointegration [44,72]. Recently, Zhang et al. constructed a Van-loaded biomimetic extracellular matrix (ECM) coating on the porous Ti. The composite coating effectively inhibited the adhesion and growth of *staphylococci* around the implant, as well as enhanced the differentiation of osteoblasts to achieve ideal osseointegration (Figure 2B) [73]. Biomimetic ECM-coating-loaded implants pointed out a novel direction for local drug delivery systems because they can promote greater tissue regeneration by simulating the microenvironment of the natural matrix.

Recently, with the development of 3D printing technology, researchers have taken advantage of its customizable materials to design novel drug delivery systems. For example, Zhang et al. prepared Van-loaded multilayer porous Ti6Al4V implants by using micro-arc oxidized technology and 3D printing technology. They confirmed that this implant could suppress infection and boost bone formation [78]. In addition, Suchý et al. constructed Van-loaded collagen/hydroxyapatite layers on the Ti-based implant via electrospun technology and 3D printing technology. They found that the composite coating could prevent the destruction of bone structure caused by bacterial infection and enhance osseointegration [79].

#### 2.2.2. Gentamicin

Gentamicin (Gent) is an aminoglycoside antibiotic. Gent has shown excellent antibacterial activity for most Gram-negative bacteria because it can block the protein synthesis of bacteria by binding to the ribosome of the virus. Thus, Gent is also often used to improve implant osseointegration by enhancing the antimicrobial capacity of implants [53,80]. For instance, Yang et al. found that Gent-loaded TNTs achieved desirable osseointegration in the rat model by significantly inhibiting the growth of bacteria and implant-associated infections [81].

It has been proved that the construction of coatings on the Ti surface has positive effects on the release and function of Gent. For instance, Sharma et al. deposited Gent-loaded silk fibroin nanoparticles coating on the Ti-based implant. They confirmed that the coating has stronger antibacterial and osteogenic properties than bare Ti [82]. Other studies suggested that Ti-based implants coated with hydroxyapatite (HA)/chitosan (Chi) composite coating had good local sustained release Gent ability, excellent biocompatibility, and osseointegration [83,84]. However, in order to extend the life of these Ti-based implants in patients, they were often necessary to add various additional antibiotics to prevent bacterial infection. To minimize the use of antibiotics, Lee et al. fabricated a heparin-based Ti implant delivery system capable of releasing Gent and BMP-2. The results showed that the system could lead to the sustained release of drugs and increase antibacterial ability. This system also significantly improved osseointegration by facilitating osteoblast activity and calcium deposition around the implant (Figure 2C) [74]. Additionally, Escobar et al. constructed a Gent-loaded Ti implant and functionalized the implant with BMP-2. The release curve of Gent met the requirements of the surgery. The implant could effectively inhibit bacterial proliferation and enhanced osseointegration [85]. These studies implied that the construction of a dual drug delivery system on Ti-based implants could solve the problem of rapid drug release and the need for a large number of additional antibiotics. In addition, researchers have prepared Gent-loaded Ti nanotubes with different pore sizes via electrochemical anodization. Previous studies showed that mesoporous biomaterials were excellent drug delivery materials because of their higher specific surface area and continuously adjustable pore sizes [86,87,88]. Draghi et al. further investigated the effect of the pore sizes of Ti-based implants on the local drug delivery system. They found that Ti nanotubes with smaller diameters performed better in terms of having antibacterial effects and improving bone–implant integration [89].

#### 2.2.3. Antimicrobial Peptides

Antimicrobial peptides (AMPs) are one kind of oligopeptide involved in immune regulation in vivo. They have excellent broad-spectrum antibacterial activity [90,91]. Therefore, AMPs can be used to enhance antimicrobial activity and bone–implant integration. For example, Kazemzadeh-Narbat et al. constructed AMP-loaded calcium phosphate coating on Ti-based implants. They found that the bone conductivity, and antibacterial and osseointegration capability of this implant were stronger than those of bare Ti [92]. To optimize the drug release ability and osseointegration of materials, researchers further fabricated different nanomorphologies on Ti substrates. For instance, Li et al. loaded AMPs on TNTs and proved that the implant had better osseointegration, with low biological toxicity, and completely inhibited the growth of bacteria [93]. In addition, Shen et al. fabricated LL37-loaded nanotubes and nanopores (NPs) on Ti substrates by the anodizing method. They provided convincing evidence that the bonding ability to Ti substrate and osteogenic differentiation capability of NPs coating was stronger. The release of LL37 significantly improved the antibacterial and osteogenic activity of the implant (Figure 2D) [75]. It is worth mentioning that the bonding strength between nanotubes and Ti substrates was poor. Thus, this study confirmed that NPs are promising candidate structures to replace nanotubes. Moreover, other studies suggested that a local drug delivery system based on mesoporous TiO_2_ implants [94] and silk fibroin/HA nanofibrous-coated Ti [95] could slowly release AMPs, avoid bone infection, and improve osseointegration. However, the bone induction activity of a single AMP-loaded implant cannot fully meet the clinical needs. Considering that, Xin et al. designed a variety of polyethylene glycol (PEG) spacer fusion peptides (FPs), including HHC36 and short peptides extracted from BMP-7 (BFP-1). A one-step reaction between the chemical groups was used to fix BFP-1 on the acid-etched Ti implants. In vivo experiments indicated that Ti implants loaded with FPs (PEG spacer no more than 12) inhibited the activity of most clinical bacteria and promoted osseointegration in rabbit models (Figure 2E) [76].

#### 2.2.4. Other Antimicrobial Drugs

Researchers fabricated several other antimicrobial drug delivery systems in anticipation of meeting clinical needs. For example, Cremer et al. prepared a porous Ti/SiO_2_ material containing the oral preservative chlorhexidine. They demonstrated that the release of chlorhexidine could facilitate osseointegration by completely preventing the formation of bacterial biofilm on the implant surface [96]. In addition, the failure of osseointegration caused by bacterial infection could be prevented by direct grafting of antibiotic ciprofloxacin on Ti implants or by loading ciprofloxacin on CS/nanoHA/Ti [97,98]. In order to obtain more superior antibacterial implants, Park et al. loaded silver nanoparticles, cephalothin, minocycline (Mino), and amoxicillin on mesoporous TiO_2_. They confirmed that the combination of silver nanoparticles and minocycline could inhibit the growth and reproduction of more kinds of bacteria [99]. Increasing attention has been paid to the construction of multifunctional local drug delivery systems by changing the coating [100,101]. For instance, Tao et al. fabricated collagen-modified Ti implants of metal–organic frameworks (MOF)@levofloxacin (Levo) coating. To achieve the effect of the slow release of Levo, gelatin (Gel) and Chi multilayers were spin-coated on the Ti implants. The composite coating could release Levo in response to pH in the bacteria-mediated, acidified microenvironment. The multifunctional coating could facilitate osseointegration by inhibiting *Escherichia coli* and *Staphylococcus aureus* and accelerating osteoblasts proliferation (Figure 2F) [77]. Additionally, Rocas et al. creatively constructed shell-stratified, amphiphilic polyurethane–polyurea (PUUa) nanoparticles on a Ti implant, and roxithromycin was wrapped in the shell. The composite coating could promote osseointegration by enhancing osteoblasts’ adhesion and suppressing bacteria growth [102]. Additionally, penicillin–streptomycin/polymer [103], tobramycin/periapatite [104], tetracycline/polymer [105]-coated Ti implants constituted drug delivery systems for local, slow-release antibiotics with superior osseointegration.

### 2.3. Anti-Bone Resorption Drug Delivery System

Osteoclasts are mainly responsible for bone resorption in the process of bone formation [106,107,108]. Therefore, better osseointegration can be ensured by anti-bone resorption drug delivery systems via mitigating osteoclast activity. Currently, bisphosphate drugs (such as alendronate (ALN), zoledronic acid (ZA), etc.), which are commonly applied to treat osteoporosis, have been proved to facilitate osseointegration of implants [109,110]. The main function of these drugs is to destroy the cytoskeleton of osteoclasts around the implant bone and inhibit the activity of osteoclasts [111,112]. Based on this property of bisphosphate drugs, researchers loaded them on the Ti implants. They demonstrated that bisphosphate drugs could inhibit the activity of osteoclasts from stimulating local bone regeneration and improve the osseointegration of Ti implants [113].

Many studies have proved that mesoporous Ti-based materials could effectively release ALN locally. For example, Pura et al. confirmed that ALN-loaded mesoporous Ti implants could slowly release ALN and had better osseointegration than bare metal [114]. In addition, Karlsson et al. analyzed the temporal and spatial distribution of drugs in ALN-loaded mesoporous TiO_2_ implants. The results showed that the drug stayed around Ti implants for a long time and promoted bone–implant integration [115,116]. Furthermore, they found that the pore size of mesoporous Ti-based materials had a great effect on the release of ALN [117]. Meanwhile, Harmankaya et al. proved that ALN-loaded mesoporous TiO_2_ implants could increase bone mineral density and enhance osseointegration in the rat tibia model [118]. In several other studies, researchers prepared different coatings on Ti-based implants to fix and slowly release ALN for better osseointegration. For instance, Guimarães et al. found that the construction of HA coating on the Ti implants could enhance the immobilization of bisphosphate drugs and bone–implant integration [119]. Additionally, Shen et al. prepared (HA-ALN/BMP-2 nanoparticle-loaded polyethylenimine (PEI)/Gel/Chi)-coated Ti6Al7Nb using the LBL technique. The multilayer membrane inhibited the growth of osteoclasts in vitro and also promoted local osseointegration of Ti6Al7Nb in the osteoporosis rabbit model in vivo (Figure 3A) [120]. The team also constructed ALN/HA/TNT and proved that the composite coating could remarkably improve osseointegration [121]. Furthermore, some researchers used ALN/HA/TNT as a nanorepository of antiosteoporosis drug raloxifene (Ral) to coordinate the regulation of osteoclasts and osteoblasts. It is worth mentioning that Ral had no side effects on the uterus and breast [97]. They found that the implant effectively decreased the activity of osteoclasts and enhanced the activity of ALP and mineralization ability of osteoblasts [122].

The release of ZA from Ti implants also showed many advantages. For example, Arnoldi et al. found that the release of ZA actively facilitated the proliferation and differentiation of mesenchymal cells and accelerated new bone formation around Ti implants in the early stage [123]. In rabbit models, ZA-loaded TNT also remarkably improved implant osseointegration and stimulated new bone formation [124]. Meanwhile, Liu et al. constructed a ZA-loaded mesoporous TiO_2_ layer (MLT-Z) on the Ti substrates. Through the slow release of ZA, the implant decreased bone resorption and promoted bone formation in vitro, and enhanced osseointegration in vivo (Figure 3B) [125]. Furthermore, the addition of HA coating, poly-D, L-lactide (PDLLA) coating, or fibroblast growth factor (bFGF) on Ti could slowly release ZA, augment the bone volume ratio, and bone binding rate [126,127]. Recently, Cui et al. filled the surface of a Ti implant with a new bisphosphate drug (technetium methylenediphosphonate (^99^Tc-MDP))-loaded poloxamer 407 hydrogel (TH/PTI). The composite scaffold stimulated the expression of genes related to osteogenic differentiation and inhibited the expression of genes related to osteoclasts. It could be clearly seen in the lower part of Figure 3C that there was an apparent gap between the bone and the Ti6Al4V scaffold, while the bone combined with the composite scaffold and grew together. Briefly, this composite scaffold could promote osseointegration in ovariectomized rabbits [128].

**Figure 3 pharmaceutics-14-01069-f003:**
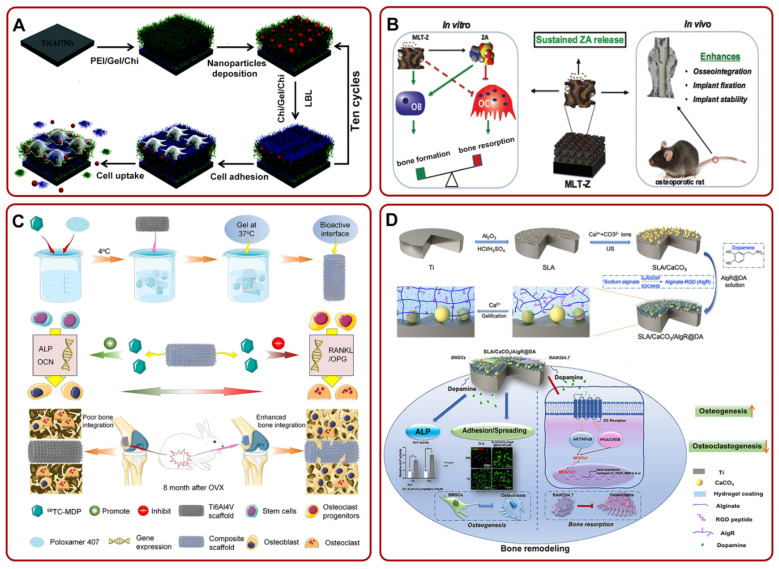
(**A**) The fabrication of Ti6Al7Nb/LBL/NP for suppressing the growth of osteoclasts and promoting local osseointegration. Adapted with permission from Ref. [120]. Copyright 2016, Royal society of chemistry; (**B**) MLT-Z coating could continue to release ZA. The release of ZA could promote bone formation and inhibit bone resorption in vitro, but also enhance osseointegration in vivo. Adapted with permission from Ref. [125]. Copyright 2018, American scientific publishers; (**C**) schematic illustration of the fabrication process of TH/PTI coating. The composite coating could enhance bone–implant integration by facilitating osteogenic differentiation and mitigating the proliferation of osteoclasts. Adapted with permission from Ref. [128]. Copyright 2021, Elsevier; (**D**) the preparation process diagram of SLA/CaCO_3_/AIgR@DA and its potential mechanism of enhancing osteogenesis and anti-bone resorption. Adapted with permission from Ref. [49]. Copyright 2021, Elsevier. Abbreviations: Chi: chitosan; Gel: gelatin; PEI: polyethylenimine; ZA: zoledronic acid; MLT-Z: ZA-loaded mesoporous TiO_2_ layer; OVX: ovariectomized.

Several other drugs have also been investigated for their ability to promote osseointegration by inhibiting osteoclast function [129]. For example, researchers fabricated calcitonin-loaded Ti alloys and proved that the local sustained release of calcitonin could suppress the activity of osteoclasts and improve osseointegration [130,131]. In addition, dopamine (DA) could coordinate osteoblasts and osteoclasts. To build a local release DA system, Wang et al. obtained rough Ti-based implants via SLA. They constructed an SLA/CaCO_3_/alginate–arginine–glycine–aspartic acid (RGD)(AlgR)@DA drug delivery system. This system could enhance bone remodeling and osseointegration by boosting the differentiation of BMSCs into osteoblasts and inhibiting the differentiation of RAW264.7 into osteoclasts (Figure 3D) [49]. A recent study suggested that KPhelligridin D could be used as a candidate drug to inhibit osteolysis and improve osseointegration of Ti-based implants [132].

### 2.4. Bone Formation Drug Delivery System

Simvastatin (SV) is a kind of drug mainly used to reduce blood lipids in the clinic [133]. Recently, it was confirmed that it also could facilitate bone formation by enhancing the expression of the bone morphogenetic protein (BMP-2) and stimulating osteoblast proliferation and differentiation [134,135]. Based on these characteristics, Yang et al. prepared porous Ti implants to release SV. They proved that SV could accelerate the proliferation and differentiation of preosteoblasts and bone–implant integration by increasing the expression of ALP, type I collagen, and osteocalcin [136]. In order to further optimize the local delivery system of SV, researchers constructed different layers on the Ti-based implants. For instance, Liu et al. loaded poly (ethylene glycol)-poly (ε-caprolactone) micelles on TNT arrays to achieve the role of cooperative slow-release SV. The implant also showed stronger osseointegration [137]. In addition, Lai et al. fabricated SV and Chi/Gel multilayer-loaded Ti-based implants. They indicated that the osseointegration capacity of Ti was improved by enhanced expression of osteogenesis-related genes and reduced osteoclast differentiation [138,139]. Moreover, SV-loaded, PLGA-coated Ti [140] and biomimetic-CaP-coated Ti alloy [141] were confirmed to have the ability to increase the survival rate of BMSCs and facilitate osseointegration of Ti. Currently, Liu et al. used 3D printing technology to add hydrogel coating to SV-loaded porous Ti. The coating could promote angiogenesis and bone regeneration, in addition to improving osseointegration [142,143]. Furthermore, to improve the bone targeting of the drug delivery system, Liu et al. grafted tetracycline (TC) in SV-loaded TNTs. TC is a widely used broad-spectrum antibiotic with a strong affinity for bone minerals. The system improved the bone targeting, antibacterial activity, and osseointegration of Ti [144].

Dexamethasone (Dex) is a kind of glucocorticoid that can promote the differentiation of BMSCs [145]. Researchers verified that Ti-based-implant-mediated drug delivery systems could achieve the continuous administration of Dex, high bioavailability, and excellent osseointegration [146,147]. On this basis, several studies optimized Ti-based implants and further explored the effect of Dex release on bone–Ti integration. For example, Yang et al. constructed the Gel/Chi multilayer-loaded TNTs. The composite layer could release Dex controllably and enhance osseointegration by promoting the proliferation and differentiation of MSCs [148]. In addition, Li et al. fabricated vertically aligned mesoporous silica thin-coating-loaded TNTs. The coating efficiently released Dex and accelerated osteogenic differentiation [149]. Recently, Wu et al. prepared polypyrrole (PPy) @HA/Dex nanocomposite coating on the Ti surface. With the release of DEX, osteogenic factors (tafazzin (TAZ), protein kinase phosphatase 1 (MKP-1), and four-and-a-half LIM domains 2 (FHL2)) were activated. They synergistically activated the osteogenic transcription factor (Runx2) and enhanced the osteogenic effect (Figure 4A) [150]. This study confirmed that PPy could effectively load Dex and promote the osteogenic ability of HA. In particular, Ran et al. constructed the silk fibroin–dexamethasone@zeolitic imidazolate framework-8 nanoparticle-loaded Ti. This Ti could control the release of DEX for a long time, enhance the expression of osteogenic related genes, and facilitate osteogenic mineralization [151]. The study suggested that the unique drug delivery system designed by the authors could also be used to deliver other osteogenic-related drugs or factors.

Several studies revealed the potential applications of antiosteoporosis drugs extracted from herbal medicines [155,156,157,158]. Icariin (ICA) is a small molecular compound extracted from traditional Chinese medicine (the Epimedium family of herbs). It can specifically facilitate bone formation and increase bone mineral density [159,160,161]. Therefore, researchers used these properties of ICA to improve bone–implant integration. For instance, Zhu et al. loaded ICA on the TNTs and confirmed that ICA-loaded TNTs could accelerate osseointegration via enhancing ECM mineralization and new bone formation. These effects were more remarkable after joining Sr [162]. Furthermore, the addition of composite coating optimized the local drug delivery system. For instance, Zhang et al. fabricated ICA-loaded TNTs and then coated them with Chi/Gel multilayer coating to seal the drug to achieve controlled release. The composite coating could stimulate the proliferation of osteoblasts and osseointegration of implant via upregulating the expression of osteoblast-related genes [163]. In addition, Ma et al. used the PLGA membrane to seal ICA on TNTs and proved that the coating could remarkably improve bone–implant integration by increasing the function of osteoblasts [164]. It is worth mentioning that the traditional surface modification methods of implants have some challenges such as expensive equipment and easy pollution. Taking this into account, Song et al. innovatively used inexpensive and clean phase-transited lysozyme (PTL) to treat the Ti surface and obtained an activated surface with high adhesion. Then, they constructed ICA-immobilized Chi/HA composite coating on PTL-primed Ti to facilitate osseointegration [165].

Several studies pointed out that some vitamins had certain bone targeting and could promote osteoblast maturation [166,167]. With these properties in mind, researchers used them to promote the osseointegration of implants. For example, He et al. used 3D printing technology to fabricate layered TNTs to simulate the trabecular structure. Then, 1α, 25-dihydroxyvitamin D3 was added to the Ti scaffold and sealed with hydrogel. The composite scaffold could facilitate early osseointegration (Figure 4B) [152]. Additionally, Sarkar et al. constructed HA-coated Ti implant and added curcumin and vitamin K2 to it. The implant loaded with dual drugs could enhance the function of osteoblasts in vitro and improve osseointegration in vivo (Figure 4C) [153].

Previous studies have shown that excessive reactive oxygen species (ROS) on the implant may inhibit the function of osteoblasts and bone–implant integration [168,169]. In order to solve this problem, Chen et al. fabricated Chi–catechol (Chi-C)/Gel/HA composite coating on the Ti substrate. The composite coating could facilitate cell adhesion and mitigate ROS damage via interfering with the expression of integrin and cadherins (Figure 4D) [154]. In addition, proanthocyanidin-loaded HA/Chi/Tiimplant constructed by Tang et al. could also effectively improve osseointegration under oxidative stress [170].

Moreover, some growth factors (e.g., transforming growth factor-β1 [171], platelet-derived growth factor-BB [172]), hormones (e.g., parathyroid hormone [173], insulin [174]) could be loaded and released by Ti-based implants to accelerate bone formation and improve osseointegration.

### 2.5. Anti-Inflammatory Drug Delivery System

As a foreign body, implants may cause an inappropriate or excessive immune response, leading to cell or tissue damage and a series of inflammatory reactions, thus resulting in poor bone–implant integration [175]. Previous studies confirmed that macrophages played an important role in regulating inflammation. Specifically, the polarization of macrophages from the M1 pro-inflammatory phenotype to the M2 anti-inflammatory phenotype could inhibit inflammation [176]. Based on this property, Shen et al. confirmed that Ti/LBL/Mino enhanced the osteogenic differentiation of mesenchymal stem cells by promoting the conversion of macrophages to anti-inflammatory phenotype (Figure 5A) [177]. In addition, the wear particles produced after the implantation of the implant were also found to cause inflammation [178,179]. Ren et al. solved the inflammation caused by wear particles around the implant by smearing erythromycin on Ti [180]. Moreover, Wei et al. fabricated poly (lactic-co-glycolic acid) (PLGA)@aspirin (ASA) nanofiber coatings on polydopamine (PDA) modified Ti via electrospinning. Studies showed that the material inhibited osteolysis caused by abrasive particles. The material also improved osseointegration and suppressed immune response (Figure 5B) [181]. On the other hand, the inadequate antioxidant capacity of cells could trigger inflammation due to the generation of excess reactive oxygen species. Inflammation may further induce poor bone–implant integration. Thus, Lee et al. loaded antioxidant N-acetyl cysteine [182] on the TNTs. The NAC-loaded TNTs could achieve local release and mitigate inflammation induced by reactive oxygen species [182,183]. In addition, indomethacin-loaded polymer-modified Ti [184], ibuprofen-loaded TNTs [185], and quercetin/CS/Ti-6Al-7Nb [186] showed excellent effects of anti-inflammation and facilitating osseointegration.

## 3. Discussion

The critical factor for successful implantation in vivo depends on excellent integration between the implant and bone. Previous studies showed that osseointegration was affected by different factors, such as bacterial infection, inflammation, etc., during the bone healing phase after implantation [25,187,188,189]. These factors may result in Ti-based implants failing to stimulate the biological activity of surrounding osteocytes, triggering infection and activating abnormal phagocytosis of macrophages. Currently, an effective strategy for enhancing osseointegration was to construct a Ti-implant-based local drug delivery system. The system could mitigate the effects of the above factors and achieve ideal osseointegration by releasing different drugs around Ti implants. For example, the release of anti-bone resorption drugs could transform bone metabolism into bone deposition by inhibiting the absorptive activity of osteoclasts [190,191]. Additionally, with the release of anti-inflammatory drugs, the system could solve aseptic loosening by mitigating local inflammatory responses [192]. Moreover, compared with the traditional systemic drug delivery, local drug delivery systems have the advantages of low dosage, low biotoxicity, high targeting, etc. Considering the above factors, the ability of antibacterial, anti-inflammatory, inhibiting bone resorption, and promoting bone formation were selected as the primary basis of this review, to help judge whether the local delivery system would enhance osseointegration.

It must be noted that although there are various ways to build drug delivery systems, they all have several limitations (Table 1). Regarding SLA, the pores only exist on the implant surface and the pore size and distribution are uncontrollable. Regarding LBL, the drug may be lost due to the less stringent processing conditions and lower bonding strength. In addition, researchers should give more consideration to the problem of coating shedding. When the coating is constructed on Ti-based implants, the relative displacement between implants and coatings gives rise to wear [193]. With the increasing range of displacement, the degree of wear is gradually deepened. This leads to the shedding of coatings.

As regards the local drug delivery system, it is very vital to ensure that the drug has a controllable release rate and continuous release time at the target site. Previously, it was a common method to construe a HA coating on the Ti-based implants to load drugs. However, the fabrication of HA coating required a high temperature. Therefore, it was difficult to load drugs well and would lead to the early release of the drugs within 1 h [194]. Currently, biodegradable coatings (such as PLGA, PDLLA, hydrogel, etc.) have widely been studied because of their controllable and continuous drug release [195]. In addition, these biodegradable coatings could also carry a greater number and variety of drugs.

Different nanostructures, such as TNTs or NPs, can be constructed on the surface of Ti substrates to release drugs directly from the surface of Ti-based implants [196]. It is worth mentioning that pharmacokinetics still needs to be taken into account. In their study, Neut et al. indicated that TNTs might lead to high biological toxicity due to explosive drug release. With that in mind, Shen et al. compared TNT structures with NP structures. The results showed that NP-loaded Ti had very low biotoxicity and could more effectively accelerate the adhesion and proliferation of osteoblasts. This study suggested that NPs may be more suitable than TNTs to improve the osseointegration of Ti-based implants. Recently, with the continuous development of the field of biomedical, an increasing number of researchers use 3D printing technology to customize the local delivery systems based on Ti-based implants [197]. However, 3D printing technology also has several challenges, such as high prices and difficulties in industrialization.

The limitation of this review lay in the small number of studies on different drugs and several variations in drug concentrations, animal models, and detection methods used in different studies. These factors may give rise to particular deviations in conclusions. In addition, most studies remained in the experimental stage. Although these local drug delivery systems showed desirable therapeutic effects in animal models, they cannot be directly applied to patients. Therefore, there is still a long way for local drug delivery systems based on Ti-based implants to be used for large-scale clinical applications.

## 4. Conclusions and Perspectives

Collectively, the construction of different local drug delivery systems, such as antimicrobial drug delivery systems, anti-bone resorption drug delivery systems, etc., on Ti-based implants can effectively improve osseointegration. However, there are still some limitations in the current methods of constructing drug delivery systems, such as infection, coating shedding, etc. Future research should improve the shortcomings of existing methods, as well as take into account the controllability and persistence of drug release. More studies should be explored to optimize local drug delivery systems, in terms of aspects such as the binding with degradable coatings, the construction of different nanostructures, and the application of new technologies. At present, it is difficult to support the transformation of Ti-based drug delivery systems to the clinical stage due to limited studies in vivo. In the future, more in vivo experiments should be carried out to promote the large-scale clinical applications of Ti-based drug delivery systems.

## Figures and Tables

**Figure 1 pharmaceutics-14-01069-f001:**
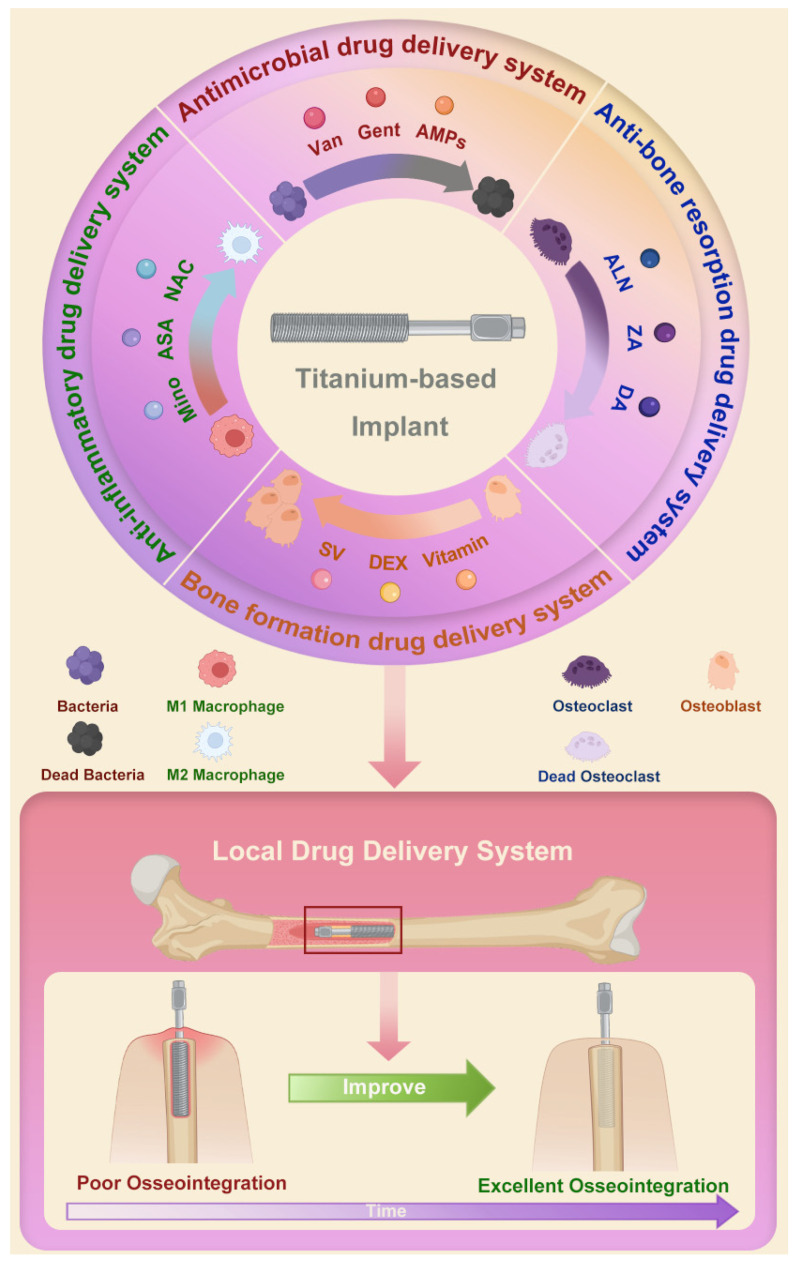
Schematic of local drug delivery systems using Ti-based implant and their role in improving osseointegration. Abbreviations: Van: vancomycin; Gent: gentamicin; AMPs: antimicrobial peptides; ALN: alendronate; ZA: zoledronic acid; DA: dopamine; SV: simvastatin; DEX: dexamethasone; Mino: minocycline; ASA: aspirin; NAC: N-acetyl cysteine.

**Figure 2 pharmaceutics-14-01069-f002:**
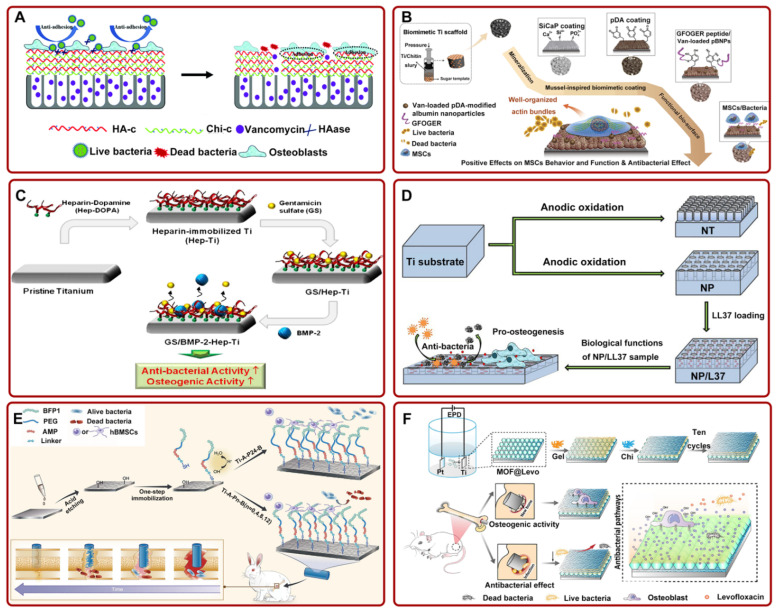
(**A**) Ti-based implants with multilayer, functional films inhibited bacterial adhesion and promoted osteoblast adhesion. Adapted with permission from Ref. [69]. Copyright 2018, Royal society of chemistry; (**B**) the preparation process diagram of Van-pBNPs/pep@pSiCaP-Ti scaffold for accelerating the osteogenic differentiation of BMSCs and inhibiting bacterial adhesion. Adapted with permission from Ref. [73]. Copyright 2020, Elsevier; (**C**) schematic illustration of the fabrication process of GS/BMP-2-Hep-Ti for enhancing antibacterial and osteogenic activity. Adapted with permission from Ref. [74]. Copyright 2012, Elsevier; (**D**) LL37-loaded NPs on Ti substrates could significantly improve the antibacterial and osteogenic activity. Adapted with permission from Ref. [75]. Copyright 2019, Dove medical press limited; (**E**) the schematic of the preparation process of FP-engineered Ti. The system could mitigate the activity of most bacteria and promote osseointegration. Adapted with permission from Ref. [76]. Copyright 2021, Elsevier; (**F**) schematic of ZIF-8@Levo-coated Ti-based implant fabrication and its antibacterial pathways. Adapted with permission from Ref. [77]. Copyright 2020, Elsevier. Abbreviations: HA-c: hyaluronate-catechol; Chi-c: chitosan–catechol; HAase: hyaluronidase; Ti: titanium; Van: vancomycin; pDA: polydopamine; MSCs: mesenchymal stem cell; SiCaP: Si-doped calcium phosphate; pBNPs: polydopamine-modified biodegradable bovine serum albumin-based nanoparticles; NT: nanotubes; NP: nanopores; PEG: polyethylene glycol; EPD: electrophoresis deposition; MOF: metal–organic framework; Levo: levofloxacin; Gel: gelatin; LBL: layer-by-layer self-assembly.

**Figure 4 pharmaceutics-14-01069-f004:**
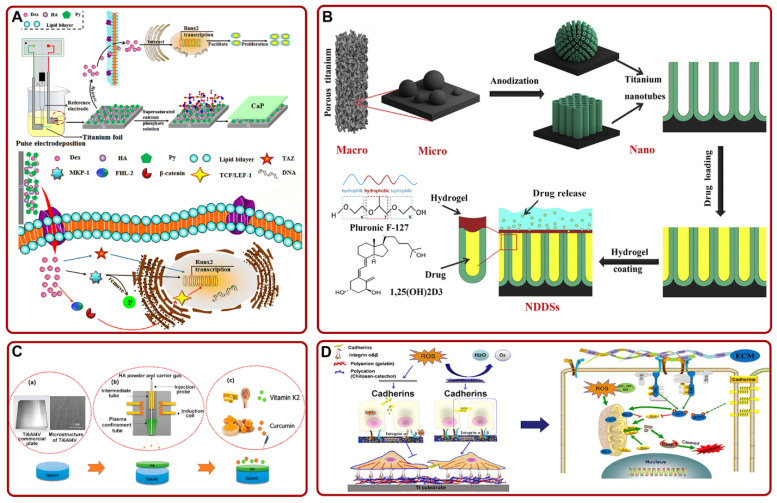
(**A**) Schematic of the mechanism that PPy@HA/Dex-composite-coated Ti promoted the osteogenic effect and osseointegration by activating Runx2. Adapted with permission from Ref. [150]. Copyright 2021, Elsevier; (**B**) a schematic diagram of constructing 1α,25-Dihydroxyvitamin D3-loaded hierarchical Ti scaffold. Adapted with permission from Ref. [152]. Copyright 2020, Elsevier; (**C**) the process of loading curcumin and vitamin K2 on HA-coated Ti6Al4V implant. Adapted with permission from Ref. [153]. Copyright 2020, American chemical society; (**D**) schematic of the proposed mechanisms of Chi-C/Gel/HA-composite-coated Ti in facilitating cell adhesion and inhibiting ROS damage. Adapted with permission from Ref. [154]. Copyright 2017, Elsevier. Abbreviations: Dex: dexamethasone; HA: hydroxyapatite; Py: pyrrole; TAZ: tafazzin; MKP-1: protein kinase phosphatase 1; FHL2: four-and-a-half LIM domains 2; TCF/LEF-1: the β-catenin binds to T-cell factor/lymphoid enhancer factor-1; NDDSs: nanoscale drug delivery systems; ROS: reactive oxygen species.

**Figure 5 pharmaceutics-14-01069-f005:**
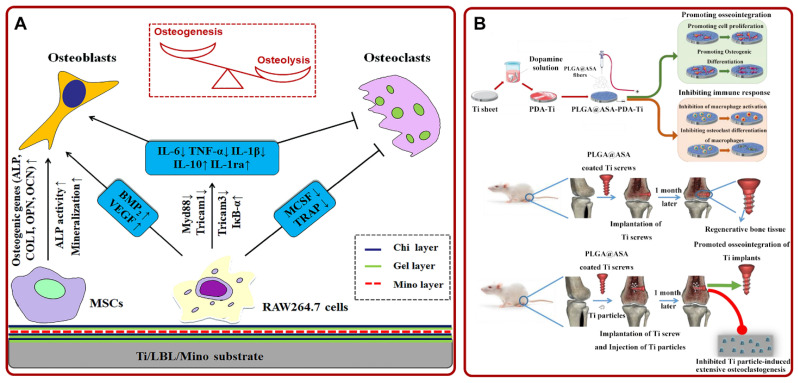
(**A**) Schematic diagram of the potential mechanism that Ti/LBL/Mino promoted osteogenesis via regulating MSCs and macrophages. Adapted with permission from Ref. [177]. Copyright 2019, Elsevier; (**B**) the preparation process and experimental diagram of PLGA@ASA-PDA-Ti. Adapted with permission from Ref. [181]. Copyright 2020, Elsevier. Abbreviations: Chi: chitosan; Gel: gelatin; Mino: minocycline; MSCs: mesenchymal stem cell; PLGA: poly (lactic-co-glycolic acid); ASA: aspirin; PDA: polydopamine; Ti: titanium.

**Table 1 pharmaceutics-14-01069-t001:** General approaches for constructing drug delivery systems using Ti-based implants.

Approach	Advantages	Limitations
Electrochemical anodization	Mature manufacturing process; good surface quality; adjustable tubular diameter	Explosive drug release; poor accuracy; contaminated electrolytic products
Sandblasting and acid etching	Enhanced hydrophilicity; large surface area; great osseointegration	Uncontrollable aperture; residual sandblasting particles; unstable roughness
Dopamine immobilization	Strong biocompatibility; excellent biodegradability; enhanced adhesion ability; wide applicability	Lower deposition rate; weak bonding strength
Layer-by-layer self-assembly	Complex coating construction; controllable coating thickness; flexible template selection; wide applicability	Poor coating stability; easy drug loss; weak bonding strength

## Data Availability

Not applicable.

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
