# Peer review of "Construction of Local Drug Delivery System on Titanium-Based Implants to Improve Osseointegration"

_pharmaceutics, 2022, doi:10.3390/pharmaceutics14051069_

Round 1
Reviewer 1 Report
The manuscript is a review on local drug delivery from titanium-based implants to improve osteointegration of implants. The review is written in a very concise form and provides an interesting overview about the published approaches on this subject. It is nicely illustrated by 5 summarizing figures, each representing a collection of graphical abstracts of selected papers on a group of released therapeutics.
Still, there are some things that need to be done.
- The authors provide many examples that base on similar surface modifications of titanium implants. A section in the beginning of this manuscript describing the general approaches to realize controlled release should be introduced. This should introduce TNT, SLA, covalent surface modifications via dopamine modification as well as LbL approaches for controlled release.
- Please also discuss to the problem of coating displacement in the discussion part.
- The figures are very detailed but small. In the PDF version, the resolution is not high enough to decipher every detail. Either increase the size of the figures or make sure that the resolution is sufficiently high. Moreover, please explain all abbreviations either in the figure legends directly or in the figure descriptions.
- Fig 3 B is not very instructive. Please either modify the figure or provide a better description.
- In Fig. 3 C, no difference in the implants of the lower part is visible that would explain the outcome.
- In Fig. 3D, the middle part shows the same things as the upper part of this Fig. Please omit this middle part and increase the size of the signaling oval.
- In Fig. 4 B, please indicate what the yellow circles represent.
Author Response
Reviewer 1:
The manuscript is a review on local drug delivery from titanium-based implants to improve osteointegration of implants. The review is written in a very concise form and provides an interesting overview about the published approaches on this subject. It is nicely illustrated by 5 summarizing figures, each representing a collection of graphical abstracts of selected papers on a group of released therapeutics.
Response: Thank you very much for the positive comments.
- The authors provide many examples that base on similar surface modifications of titanium implants. A section in the beginning of this manuscript describing the general approaches to realize controlled release should be introduced. This should introduce TNT, SLA, covalent surface modifications via dopamine modification as well as LbL approaches for controlled release.
Response: Thank you very much for the professional comments and valuable suggestions. According to your suggestions, the ‘2.1 Construction approaches of local drug delivery systems on Ti-based implants’ section was added and highlighted in the revised manuscript. (Page 6, Line 8-9; Page 7, Line 1-29; Page 8, Line 1-3)
2.1 Construction approaches of local drug delivery systems on Ti-based implants
In order to realize controlled drug delivery, it is necessary to fabricate appropriate drug delivery systems on Ti-based implants. At present, the main approaches of constructing drug delivery systems on Ti-based implants include electrochemical anodization, sandblasting and acid etching (SLA), dopamine (DA) immobilization, and layer-by-layer (LBL) self-assembly. The advantages and limitations of the above methods are presented in Table 1.
Electrochemical anodization: Electrochemical anodization is a strategy of forming an oxide film on the surface of metals and their alloys [39,40]. This method is usually used to fabricate TiO2 nanotubes (TNTs) when constructing a drug delivery system [41]. TNTs are arranged vertically on Ti substrates to simulate the nanostructures in natural tissues [42]. On the one hand, the tubular diameters of TNT can be adjusted by changing the voltage and pH in the anodizing process to obtain a suitable tubular structure to load and deliver drugs [43]. On the other hand, TNTs prepared by anodization have been proved to regulate the behavior of osteoblasts and stem cells and effectively improve osseointegration [44].
SLA: Currently, SLA is the most commonly used strategy for surface modification of implants [45]. This strategy means that the abrasive medium material is sprayed on the surface of the implant by high-speed air flow to form a depression [46,47]. After that, acid etching is used to form smaller secondary structures and to clean impurities on the implant surface [48]. SLA can increase the roughness of implants, facilitate drugs loading and accelerate new bone formation around the implant [49].
DA immobilization: DA immobilization refers to the loading of drugs or factors on the Ti-based implants with the assistance of DA [50]. On the one hand, the chemical composition of DA is similar to that of mussel adhesion proteins, which have strong adhesion and can stabilize drugs or other bioactive molecules [51]. On the other hand, DA has excellent biocompatibility and biodegradability in vivo [52].
LBL self-assembly: LBL self-assembly is a surface modification method based on the alternating assembly of oppositely charged polyelectrolytes to fabricating multi-layer coatings [53,54]. This method is not only easy to control the thickness of coatings, but also can release drugs layer by layer to promote osseointegration [55].
Table 1. General approaches for constructing drug delivery systems on Ti-based implants
|
Approach |
Advantages |
Limitations |
|
Electrochemical anodization |
Mature manufacturing process; good surface quality; adjustable tubular diameter |
Explosive drug release; poor accuracy; contaminated electrolytic products |
|
Sandblasting and acid etching |
Enhanced hydrophilicity; large surface area; great osseointegration |
Uncontrollable aperture; residual sandblasting particles; unstable roughness |
|
Dopamine immobilization |
Strong biocompatibility; excellent biodegradability; enhanced adhesion ability; wide applicability |
Lower deposition rate; weak bonding strength |
|
Layer-by-layer self-assembly |
Complex coating construction; controllable coating thickness; flexible template selection; wide applicability |
Poor coating stability; easy drug loss; weak bonding strength |
- Please also discuss to the problem of coating displacement in the discussion part.
Response: Thank you very much for your professional and constructive suggestions. According to your suggestions, the discussion part was revised and highlighted in the revised manuscript. (Page 23, Line 14-22)
It must be noted that although there are various ways to build drug delivery systems, they all have several limitations (Table1). For SLA, the pores only exist on the implant surface and the pore size and distribution are uncontrollable. For LBL, the drug may be lost due to the less stringent processing conditions and lower bonding strength. In addition, researchers should pay more attention to the problem of coating shedding. When the coating is constructed on Ti-based implants, the relative displacement between implants and coatings gives rise to wear [193]. With the increasing range of displacement, the degree of wear is gradually deepened. This leads to the shedding of coatings.
- The figures are very detailed but small. In the PDF version, the resolution is not high enough to decipher every detail. Either increase the size of the figures or make sure that the resolution is sufficiently high. Moreover, please explain all abbreviations either in the figure legends directly or in the figure descriptions.
Response: Thank you very much for your careful observations and constructive suggestions. According to your suggestion, the figures and figure descriptions were revised in the revised manuscript. All authors hope that the revision will meet with approval. (Page 6, Line 3-5; Page 10, Line 10-15; Page 16, Line 19-11; Page 17, Line 1-2; Page 19, Line 10-13; Page 22, Line 11-13)
Abbreviations: Van: Vancomycin; Gent: Gentamicin; AMPs: Antimicrobial peptides; ALN: Alendronate; ZA: Zoledronic acid; DA: Dopamine; SV: Simvastatin; DEX: Dexamethasone; Mino: Minocycline; ASA: Aspirin; NAC: N-acetyl cysteine.
Abbreviations: HA-c: hyaluronate-catechol; Chi-c: chitosan-catechol; HAase: Hyaluronidase; Ti: titanium; Van: Vancomycin; pDA: polydopamine; MSCs: mesenchyma stem cell; SiCaP: Si-doped calcium phosphate; pBNPs: polydopamine-modified biodegradable bovine serum albumin-based nanoparticles; NT: nanotubes; NP: nanopores; PEG: polyethylene glycol; EPD: electrophoresis deposition; MOF: metal-organic framework; Levo: levofloxacin; Gel: gelatin; LBL: layer-by-layer self-assembly.
Abbreviations: Chi: Chitosan; Gel: gelatin; PEI: polyethylenimine; ZA: Zoledronic acid; MLT-Z: ZA loaded-mesoporous TiO2 layer; OVX: ovariectomized.
Abbreviations: Dex: Dexamethasone; HA: hydroxyapatite; Py: pyrrole; TAZ: tafazzin; MKP-1: protein kinase phosphatase 1; FHL2: four and a half LIM domains 2; TCF/LEF-1: the β-catenin binds to T cell factor/lymphoid enhancer factor-1; NDDSs: nanoscale drug delivery systems; ROS: reactive oxygen species.
Abbreviations: Chi: Chitosan; Gel: gelatin; Mino: Minocycline; MSCs: mesenchymal stem cell; PLGA: poly (lactic-co-glycolic acid); ASA: aspirin; PDA: polydopamine; Ti: titanium.
- Fig. 3B is not very instructive. Please either modify the figure or provide a better description.
Response: Thank you very much for the kind suggestion. According to your suggestion, the description was revised in the revised manuscript. (Page 16, Line 13-15)
MLT-Z coating could continue to release ZA. The release of ZA could not only promote bone formation and inhibit bone resorption in vitro, but also enhance osseointegration in vivo [125].
- In Fig. 3C, no difference in the implants of the lower part is visible that would explain the outcome.
Response: Thank you very much for the careful observation. According to your suggestion, the description was explained in the revised manuscript. (Page 16, Line 6-10)
It could be clearly seen in the lower part of Figure 3C that there was an apparent gap between the bone and the Ti6Al4V scaffold, while the bone combined with composite scaffold and grew together. Briefly, this composite scaffold could promote osseointegration in ovariectomized rabbits [128].
- In Fig. 3D, the middle part shows the same things as the upper part of this Fig. Please omit this middle part and increase the size of the signaling oval.
Response: Thank you very much for the kind suggestion. The Fig. 3D was revised in the revised manuscript.
- In Fig. 4B, please indicate what the yellow circles represent.
Response: Thank you very much for the good suggestion. The yellow circles represent “Drug release”. This was explained in Fig. 4B in the revised manuscript.

Reviewer 2 Report
In this manuscript, the authors reviewed some local drug delivery systems on titanium-based implants to improve osseointegration. In my opinion, some issues should be further addressed and I hope the following comments could be helpful for improving their paper.
-The authors should summarize the current approaches of "local drug delivery systems on titanium-based implants" and compare their advantages and disadvantages in order to draw the reader's attention. It is better to add a table.
- Good quality figures are very important for a good review paper. The quality of the figures should be improved, please.
-The author should improve the conclusion of the manuscript, and insert the futures perspectives about the subject.
-Please draw a graphical abstract to draw the reader's attention.
Author Response
Reviewer 2:
In this manuscript, the authors reviewed some local drug delivery systems on titanium-based implants to improve osseointegration. In my opinion, some issues should be further addressed and I hope the following comments could be helpful for improving their paper.
Response: Thank you very much for the positive comments and kind suggestions.
- The authors should summarize the current approaches of "local drug delivery systems on titanium-based implants" and compare their advantages and disadvantages in order to draw the reader's attention. It is better to add a table.
Response: Thank you very much for the professional comments and valuable suggestions. According to your suggestions, the ‘2.1 Construction approaches of local drug delivery systems on Ti-based implants’ section was added and highlighted in the revised manuscript. (Page 6, Line 8-9; Page 7, Line 1-29; Page 8, Line 1-3)
2.1 Construction approaches of local drug delivery systems on Ti-based implants
In order to realize controlled drug delivery, it is necessary to fabricate appropriate drug delivery systems on Ti-based implants. At present, the main approaches of constructing drug delivery systems on Ti-based implants include electrochemical anodization, sandblasting and acid etching (SLA), dopamine (DA) immobilization, and layer-by-layer (LBL) self-assembly. The advantages and limitations of the above methods are presented in Table 1.
Electrochemical anodization: Electrochemical anodization is a strategy of forming an oxide film on the surface of metals and their alloys [39,40]. This method is usually used to fabricate TiO2 nanotubes (TNTs) when constructing a drug delivery system [41]. TNTs are arranged vertically on Ti substrates to simulate the nanostructures in natural tissues [42]. On the one hand, the tubular diameters of TNT can be adjusted by changing the voltage and pH in the anodizing process to obtain a suitable tubular structure to load and deliver drugs [43]. On the other hand, TNTs prepared by anodization have been proved to regulate the behavior of osteoblasts and stem cells and effectively improve osseointegration [44].
SLA: Currently, SLA is the most commonly used strategy for surface modification of implants [45]. This strategy means that the abrasive medium material is sprayed on the surface of the implant by high-speed air flow to form a depression [46,47]. After that, acid etching is used to form smaller secondary structures and to clean impurities on the implant surface [48]. SLA can increase the roughness of implants, facilitate drugs loading and accelerate new bone formation around the implant [49].
DA immobilization: DA immobilization refers to the loading of drugs or factors on the Ti-based implants with the assistance of DA [50]. On the one hand, the chemical composition of DA is similar to that of mussel adhesion proteins, which have strong adhesion and can stabilize drugs or other bioactive molecules [51]. On the other hand, DA has excellent biocompatibility and biodegradability in vivo [52].
LBL self-assembly: LBL self-assembly is a surface modification method based on the alternating assembly of oppositely charged polyelectrolytes to fabricating multi-layer coatings [53,54]. This method is not only easy to control the thickness of coatings, but also can release drugs layer by layer to promote osseointegration [55].
Table 1. General approaches for constructing drug delivery systems on Ti-based implants
|
Approach |
Advantages |
Limitations |
|
Electrochemical anodization |
Mature manufacturing process; good surface quality; adjustable tubular diameter |
Explosive drug release; poor accuracy; contaminated electrolytic products |
|
Sandblasting and acid etching |
Enhanced hydrophilicity; large surface area; great osseointegration |
Uncontrollable aperture; residual sandblasting particles; unstable roughness |
|
Dopamine immobilization |
Strong biocompatibility; excellent biodegradability; enhanced adhesion ability; wide applicability |
Lower deposition rate; weak bonding strength |
|
Layer-by-layer self-assembly |
Complex coating construction; controllable coating thickness; flexible template selection; wide applicability |
Poor coating stability; easy drug loss; weak bonding strength |
- Good quality figures are very important for a good review paper. The quality of the figures should be improved, please.
Response: Thank you very much for your careful observations and constructive suggestions. According to your suggestion, the figures were revised in the revised manuscript. All authors hope that the revision will meet with approval.
- The author should improve the conclusion of the manuscript, and insert the futures perspectives about the subject.
Response: Thank you very much for your professional and constructive suggestion. According to your suggestions, the conclusion section was revised in the revised manuscript. (Page 24, Line 27-29; Page 25, Line 1-8)
However, there are still some limitations in the current methods of constructing drug delivery systems, such as infection, coating shedding, and so on. Future research should not only improve the shortcomings of existing methods, but also take into account the controllability and persistence of drug release. More studies should be explored to optimize local drug delivery systems, such as the binding with degradable coatings, the construction of different nanostructures and the application of new technologies. At present, it is difficult to support the transformation of Ti-based drug delivery systems to the clinical stage due to limited studies in vivo. In the future, more in vivo experiments should be carried out to promote the large-scale clinical applications of Ti-based drug delivery systems.
- Please draw a graphical abstract to draw the reader's attention.
Response: Thank you very much for your valuable comments. The graphical abstract was provided in the revised manuscript.

Round 2
Reviewer 2 Report
Accept in present form.